# A Case Report of Advanced Pulmonary Adenocarcinoma in a Dog Managed with Chemotherapy and Cytokine-Based Immunotherapy

**DOI:** 10.3390/ani15223330

**Published:** 2025-11-19

**Authors:** Kyu-Duk Yeon, Ji-Hyuk Seo, Jung-Hyun Kim

**Affiliations:** 1Department of Veterinary Internal Medicine, College of Veterinary Medicine, Konkuk University, Seoul 05029, Republic of Korea; gundam223@naver.com; 2SNC Animal Medical Center, 416 Nonhyeon-ro, Gangnam-gu, Seoul 06134, Republic of Korea; nurong3000@hanmail.net

**Keywords:** dog, primary pulmonary adenocarcinoma, NK cell immunotherapy, vinorelbine, immunotherapy

## Abstract

**Simple Summary:**

Pulmonary adenocarcinoma in dogs, particularly in advanced stages, carries a poor prognosis and limited treatment options. This report describes the clinical course of a dog with advanced pulmonary adenocarcinoma diagnosed based solely on cytology, without lymph node cytology to confirm nodal metastasis. The dog initially achieved temporary disease stabilization following cytokine-based immunotherapy after vinorelbine-associated adverse events. However, disease progression later occurred with the development of a metastatic lesion in the contralateral lung, prompting resumption of combination therapy. The dog survived for 241 days in total, including 143 days after progression to stage IV disease, exceeding previously reported survival durations for similar cases. These findings suggest that combining chemotherapy with cytokine-based immunotherapy may provide clinical benefit in advanced, non-surgical pulmonary carcinoma in dogs.

**Abstract:**

Pulmonary adenocarcinoma in dogs, particularly in advanced stages, carries a poor prognosis with limited therapeutic options. Immunotherapeutic approaches that activate natural killer (NK) cells may provide additional clinical benefit. This report describes the clinical response and survival outcome of a 9-year-old neutered male Welsh Corgi with late-stage pulmonary adenocarcinoma treated with combined chemotherapy and cytokine-based NK cell-activating immunotherapy. The dog presented with intermittent coughing, dyspnea, and cyanosis. Imaging revealed a large pulmonary mass with suspected nodal metastasis (stage III, T4N1M0). Cytology confirmed pulmonary adenocarcinoma. A splenic myelolipoma, unrelated to the primary pulmonary tumor, was identified incidentally and surgically removed. Treatment included vinorelbine-based chemotherapy and cytokine-based immunotherapy using interleukin (IL)-15, IL-12, IL-23, and selenium. After temporary discontinuation due to adverse events, cytokine monotherapy was administered, followed by resumed combination therapy upon stage IV progression with contralateral lung metastasis. Radiographic follow-up demonstrated disease stabilization during monotherapy and prolonged survival with combination therapy. The dog survived for 241 days, including 143 days after stage IV diagnosis, exceeding previously reported outcomes. Although NK cell function was not directly evaluated, these findings raise the possibility that cytokine-based NK cell immunotherapy, when combined with chemotherapy, could have contributed to disease control and prolonged survival in advanced canine pulmonary adenocarcinoma.

## 1. Introduction

Pulmonary adenocarcinoma (PAC) is the most common primary lung tumor in dogs, though it remains relatively uncommon overall in small-animal practice [1]. Most affected dogs are middle-aged to older, and early-stage disease often goes unnoticed because clinical signs such as cough, exercise intolerance, or dyspnea usually develop only after the tumor has progressed. As a result, many cases are diagnosed at an advanced stage, when curative surgery is no longer feasible. Clinical stage strongly influences prognosis, with median survival times reported as 26–60 days in dogs with lymph node involvement and 52 days for surgically managed stage IV disease, compared with over 900 days in stage I cases [1,2].

Surgical lung lobectomy is the treatment of choice for localized PAC; however, the majority of dogs with advanced or metastatic disease are not candidates for surgery. In such cases, systemic chemotherapy—including vinorelbine, platinum-based agents, or metronomic protocols—may provide palliative benefit, but survival outcomes remain limited [3,4]. These challenges have driven growing interest in integrating novel adjunctive therapies with conventional approaches.

In human oncology, immunotherapy has significantly changed the treatment landscape by harnessing the immune system against cancer. Natural killer (NK) cell-based strategies are particularly attractive because NK cells can eliminate malignant cells without prior antigen sensitization [5]. Cytokines such as interleukin (IL)-12 and IL-15 enhance NK cell proliferation and antitumor activity, and IL-15 in particular has gained attention as a promising NK-activating agent [6,7]. In veterinary medicine, clinical evidence remains limited, although recent exploratory studies—including aerosolized IL-15 therapy in dogs with pulmonary metastases—have suggested feasibility and potential benefit [8]. Moreover, pulmonary adenocarcinoma has been a key focus of NK cell–based immunotherapy research in human lung cancer, underscoring its translational relevance as a target for similar strategies in dogs [9].

Given the poor prognosis of advanced PAC and the limited efficacy of chemotherapy alone, combining immune-modulating treatments with standard therapy may offer improved clinical outcomes. This case report describes the progression and management of a dog with cytology-confirmed PAC treated with vinorelbine-based chemotherapy and cytokine-based NK-activating immunotherapy, highlighting the potential value of a multimodal approach for advanced, non-surgical PAC.

## 2. Case Description

A 9-year and 8-month-old neutered male Welsh corgi weighing 13.5 kg was referred to our hospital after the incidental detection of pulmonary and splenic masses during routine health screening. At the time of presentation, the dog was clinically stable. However, the medical history included intermittent coughing, dyspnea, and cyanosis. Thoracic CT revealed an 81.3 mm pulmonary mass in the right caudal lung lobe extending into the accessory lobe. The middle tracheobronchial lymph node was enlarged and showed strong rim contrast enhancement, a CT feature commonly associated with metastatic lymph node involvement due to peripheral neoplastic infiltration and increased vascularization, making metastasis highly suspected (Figure 1). A 6.5 × 4.7 × 5.3 cm splenic mass was also identified.

Surgical resection and histopathological evaluation of both lesions were recommended. However, because of the owner’s concerns regarding anesthetic risk and potential complications, thoracic surgery was declined. Instead, the pulmonary mass was evaluated cytologically using fine-needle aspiration (FNA) (Figure 2). Cytology revealed clusters of ovals to polygonal epithelial cells arranged in dense acinar formations with indistinct cell borders and intercellular junctions. The cells exhibited mild to moderate anisocytosis, moderate to high nucleus-to-cytoplasm ratios, oval nuclei with one to two prominent nucleoli, and frequent mitotic figures. These findings were consistent with epithelial neoplasia, most compatible with pulmonary adenocarcinoma. The splenic mass was surgically excised at week 4 and was diagnosed as a benign myelolipoma. No evidence of an extrapulmonary primary tumor was identified on imaging. Based on cytologic findings, imaging characteristics, and the absence of other primary lesions, the pulmonary lesion was diagnosed as a primary pulmonary adenocarcinoma (stage III, T4N1M0).

The timeline of clinical events and treatments is summarized in Figure 3. Adverse events (AEs) and disease progression were assessed at baseline and during follow-up evaluations performed every one to two weeks. AEs were graded according to the Veterinary Cooperative Oncology Group Common Terminology Criteria for Adverse Events (VCOG-CTCAE v2.0) [10], and tumor response and disease progression were evaluated using the Response Evaluation Criteria in Solid Tumors (RECIST) [11] based on thoracic radiographs. 

At week 6, palliative treatment was initiated with vinorelbine (15 mg/m^2^ IV; Navelbine^®^, Pierre Fabre, Boulogne, France), piroxicam (0.3 mg/kg PO once daily), and cytokine-based NK cell-activator immunotherapy, which included IL-15, IL-12, IL-23, and selenium. Each cytokine was prepared at a concentration of 0.1 mg/mL, and selenium at 0.25 mg/mL, in accordance with the manufacturer’s instructions. A total volume of 10 mL of the cytokine formulation was administered. The cytokine dose and schedule were administered according to the manufacturer’s recommended protocol, selected within the safety ranges reported for cytokine-based immunotherapy in veterinary and translational studies. This approach was informed by previously reported safe systemic use of IL-based immunotherapy in dogs and translational research.

Four days after pulmonary FNA and treatment initiation, thoracic radiography was performed due to acute respiratory distress. Compared with the pretreatment images, follow-up radiographs showed pneumothorax, increased alveolar opacity, and mild pleural effusion (Figure 4).

Hematologic analysis indicated grade 4 neutropenia (neutrophil count: 0.23 K/μL) and an elevated C-reactive protein (8.3 mg/dL). The dog was hospitalized for approximately one week and received oxygen supplementation, granulocyte colony-stimulating factor, and meropenem (Meropen^®^, Daewoong Pharmaceutical Co., Seoul, Republic of Korea), with subsequent clinical improvement.

At the owner’s request, vinorelbine was discontinued from week 8 to week 13, and cytokine therapy was continued as monotherapy. During this period, cytokine therapy was administered weekly in five consecutive doses. The dog remained clinically stable without any significant symptoms, and thoracic radiography revealed stable disease.

Routine follow-up thoracic radiographs at week 14 revealed a new metastatic lesion in the contralateral lung (Figure 5A), prompting reclassification to stage IV disease and resumption of combination therapy. NK cell immunotherapy was co-administered with each chemotherapy session. Vinorelbine (Navelbine^®^, Pierre Fabre, Boulogne, France) was re-administered at 7.5 mg/m^2^ IV every 10 days for four cycles, followed by 15 mg/m^2^ every 14 days for an additional three cycles. Stable disease was maintained for approximately 6 weeks (from week 24 to week 30; Figure 5B), although grade 1 anorexia developed after the dose was increased.

By week 30, imaging confirmed disease progression (Figure 5C), and vinorelbine was replaced with carboplatin (200 mg/m^2^ IV every three weeks; Carbotinol^®^, Korea United Pharm Inc., Seoul, Republic of Korea) while piroxicam and cytokine therapy were administered.

At 34 weeks, the dog developed acute respiratory failure. Imaging revealed extensive tumor-associated emphysema and compression of the contralateral lung (Figure 5D). The patient’s condition rapidly deteriorated, and death occurred later that day.

The total volume of the cytokine formulation administered throughout the course of treatment was 10 mL, prepared at the specified concentrations. No further proprietary details regarding the formulation are disclosed.

## 3. Discussion

This case report describes a dog with advanced-stage primary pulmonary adenocarcinoma, a relatively uncommon diagnosis in veterinary patients, which demonstrated prolonged survival following a multimodal treatment approach. Although vinorelbine- and cytokine-based NK cell immunotherapy was initially interrupted owing to adverse events, temporary stabilization was achieved with cytokine monotherapy, allowing for the resumption of combination therapy. Notably, the dog achieved a total survival time of approximately 35 weeks, including approximately 20 weeks after stage IV progression, which substantially exceeded previously reported median survival durations in comparable cases. These findings suggest that a tailored combination of standard chemotherapy and immunotherapy may offer meaningful clinical benefits, even in dogs with advanced unresectable pulmonary carcinoma.

Pulmonary adenocarcinoma in dogs, particularly stage IV, with nodal or distant metastasis, is generally associated with a poor prognosis. Previous studies have reported a median survival time of approximately 26–60 days in dogs with nodal involvement, and only 52 days in dogs with stage IV disease undergoing surgical treatment, in contrast to over 900 days in dogs with stage I disease [1,2]. Vinorelbine monotherapy has demonstrated limited efficacy in dogs with advanced-stage disease, with a median survival of 100 days, whereas metronomic protocols yield modest outcomes [3,4]. In contrast, the present patient showed a total survival of approximately 35 weeks, including approximately 20 weeks after stage IV diagnosis, substantially exceeding previously reported outcomes, suggesting a meaningful clinical benefit.

In the present case, pneumothorax, pneumonia, and grade 4 neutropenia developed shortly after pulmonary FNA and initial administration of vinorelbine. Although these complications may develop independently, their close temporal association raises the possibility of a cumulative effect involving FNA-induced lung injury and chemotherapy-related immunosuppression. Notably, grade 4 neutropenia occurred only after the first chemotherapy dose, which was administered during hospitalization with daily blood tests, and was not observed during subsequent outpatient cycles at the same dosage. This suggests that the initial findings may have been incidental and were detected because of more intensive monitoring. These observations highlight the importance of carefully timing invasive diagnostic procedures and cytotoxic chemotherapy, particularly in patients at risk of compromised pulmonary function or immune status.

Percutaneous fine-needle aspiration (FNA) is a commonly used diagnostic method for pulmonary tumors with generally low complication rates. Pneumothorax is the most frequent adverse event, and although often self-limiting, it may be clinically significant in patients with compromised pulmonary function [12]. Infectious complications have also been reported, particularly in immunocompromised patients [13]. These observations highlight the need for careful monitoring when combining invasive procedures with immunosuppressive treatments.

Paratumoral emphysema, although rare, has been sporadically reported in canine pulmonary adenocarcinoma [14,15]. While its prognostic impact remains unclear [16], it may exacerbate respiratory compromise. Awareness of this finding is important when interpreting radiographic changes and evaluating clinical progression.

Interleukin-15 (IL-15) and interleukin-12 (IL-12) are well-established NK cell-activating cytokines that promote antitumor immunity by enhancing NK cell proliferation, cytotoxicity, and IFN-γ secretion [6,17]. IL-15 plays a critical role in the survival and homeostasis of CD8+ T cells and NK cells [6], while IL-12 shifts the tumor microenvironment toward an immunostimulatory state through IFN-γ induction [7]. Cytokine stimulation has also been shown to generate cytokine-induced memory-like (CIML) NK cells that exhibit sustained antitumor responses upon re-exposure [17,18]. In this case, although IL-18 was not included, IL-23 may have functioned as a co-stimulatory factor that synergizes with IL-12 and IL-15 to promote NK cell activation and IFN-γ secretion [19]. Selenium may further support NK cell activity and modulate the tumor microenvironment via antioxidant and metabolic pathways [20].

Although the clinical data on cytokine-based NK cell immunotherapy are increasing in human oncology [5], their use in veterinary oncology remains limited. A few recent cases, such as those involving inhaled IL-15 therapy in canine lung cancer, have shown promising outcomes [8]. However, combination strategies integrating NK cell-targeted cytokines with conventional chemotherapy have rarely been reported in dogs. This study was designed to explore the feasibility and therapeutic potential of combining IL-12, IL-15, IL-23, and selenium with chemotherapy to activate NK cells and enhance their antitumor effects. 

Here, cytokine monotherapy may have contributed to temporary disease stabilization and symptom relief, which is consistent with previous findings in epithelial tumors [6]. Upon combination with vinorelbine, multiple immunologic synergies may have supported therapeutic efficacy: (1) IL-15 may have counteracted chemotherapy-induced lymphodepletion, preserving immune function [6]; (2) tumor cell injury from chemotherapy may have increased susceptibility to NK-mediated cytotoxicity, amplified by IL-12/IL-15 activation [7]; and (3) the combination may have promoted immunogenic cell death, tumor antigen release, and favorable toxicity profiles via complementary mechanisms [18,19,20]. Although NK cell activity was not directly assessed, prior studies support that IL-12, IL-15, and IL-23 enhance NK cell proliferation, IFN-γ production, and antitumor cytotoxicity [6,18,19]. The prolonged survival observed in this patient might be partly attributable to NK cell–mediated antitumor activity, although direct evidence was not obtained.

This case report has several limitations. First, a definitive histopathological diagnosis and tumor grading were not obtained because thoracic surgery was declined, and only cytology was performed. Therefore, although cytologic features were consistent with PAC, histologic confirmation and grading—which are important prognostic indicators—were not available. Second, metastasis to the tracheobronchial lymph node was strongly suspected based on CT findings; however, cytology or histopathology was not performed, and reactive lymphadenopathy cannot be fully excluded. These factors limit the accuracy of staging and outcome interpretation.

Beyond the uncertainty in staging accuracy, an additional limitation relates to immune monitoring. The absence of direct measurement of NK cell activity or cytokine-induced immune biomarkers (e.g., IFN-γ, perforin, and granzyme B) precludes definitive conclusions regarding the immunologic effects of the therapy. However, previous studies have shown that stimulation with IL-12 and IL-15 can induce memory-like NK cells with enhanced cytotoxic activity [17,18]. Future investigations incorporating quantitative immune profiling, cytokine pharmacokinetics, and appropriate control comparisons are warranted to validate these observations and clarify the therapeutic potential of NK cell activation in veterinary oncology.

This case highlights the feasibility and clinical potential of combining cytokine-based NK cell stimulation with conventional chemotherapy for the treatment of advanced canine pulmonary carcinoma. Future prospective studies with larger sample sizes, immune monitoring, and standardized treatment protocols are essential to establish evidence-based guidelines for immune-chemotherapy in veterinary cancer care.

## 4. Conclusions

This case demonstrated the potential clinical benefits of combining vinorelbine chemotherapy with NK cell-activating cytokines (IL-15, IL-12, IL-23, and selenium) in a dog with stage IV PAC. Although the immune function was not directly assessed, the observed disease stabilization and survival extension suggested a potential contribution of NK cell-mediated antitumor effects. The cytokine formulation used in this case (0.1 mg/mL for each cytokine and 0.25 mg/mL for selenium; total volume, 10 mL) may serve as a reference for future investigations. Further studies are needed to evaluate the immune responses, refine dosing strategies, and explore the broader utility of this approach in canine oncology. 

## Figures and Tables

**Figure 1 animals-15-03330-f001:**
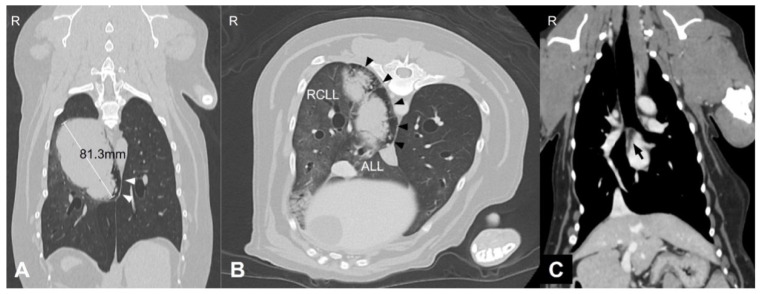
Thoracic computed tomography (CT) images of the lung mass. (**A**) Transverse image showing a large soft-tissue attenuating mass (arrowhead) in the right caudal lung lobe (RCLL) with mild peritumoral emphysematous changes along the accessory lung lobe (ALL) dorsomedial margin. (**B**) Dorsal plane image showing the maximum diameter of the mass (arrowhead) measuring 81.3 mm. (**C**) Soft-tissue window image showing enlargement of the middle tracheobronchial lymph node (arrow) located caudal to the carina. Abbreviations: RCLL, right caudal lung lobe; ALL, accessory lung lobe; R, right side.

**Figure 2 animals-15-03330-f002:**
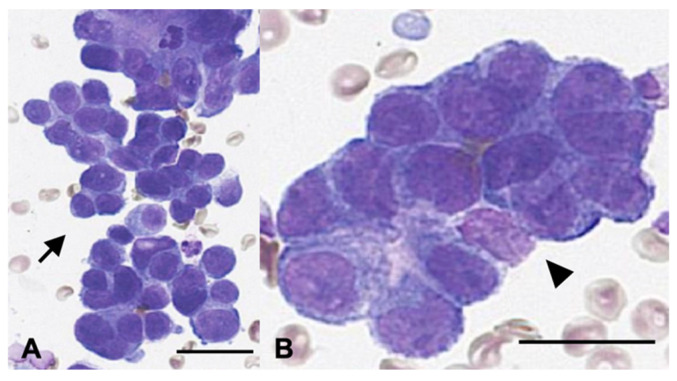
Diff-Quik-stained cytology smear obtained from the pulmonary mass. (**A**) Low-power view showing clusters of epithelial cells arranged in an acinar pattern (arrow). Scale bar = 50 µm. (**B**) High-power view highlighting pleomorphic epithelial cells (arrowhead). Scale bar = 20 µm.

**Figure 3 animals-15-03330-f003:**
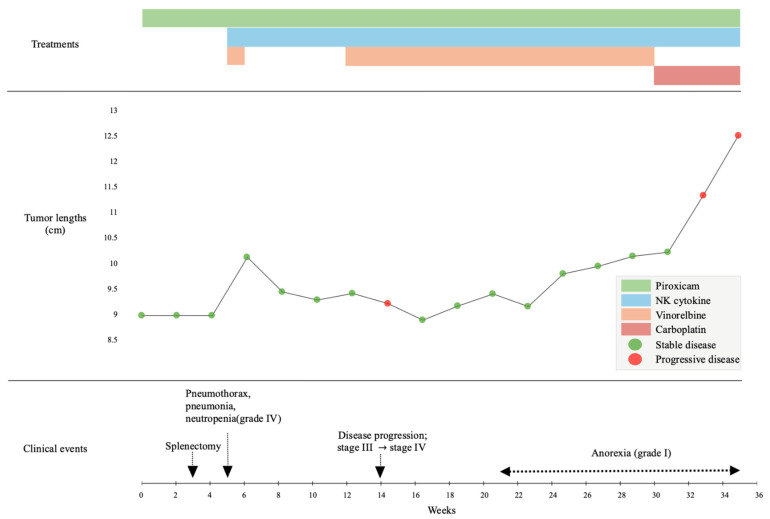
Timeline of clinical events, treatments, and tumor responses over a 36-week course. Vinorelbine (orange), piroxicam (green), and cytokine-based NK cell immunotherapy (blue) were administered according to the schedule shown. Carboplatin (red) was initiated upon disease progression. Tumor response was evaluated using RECIST-like criteria (stable disease, green; progressive disease, red). Key clinical events, including splenectomy, hospitalization, and onset of anorexia, are indicated by dotted arrows.

**Figure 4 animals-15-03330-f004:**
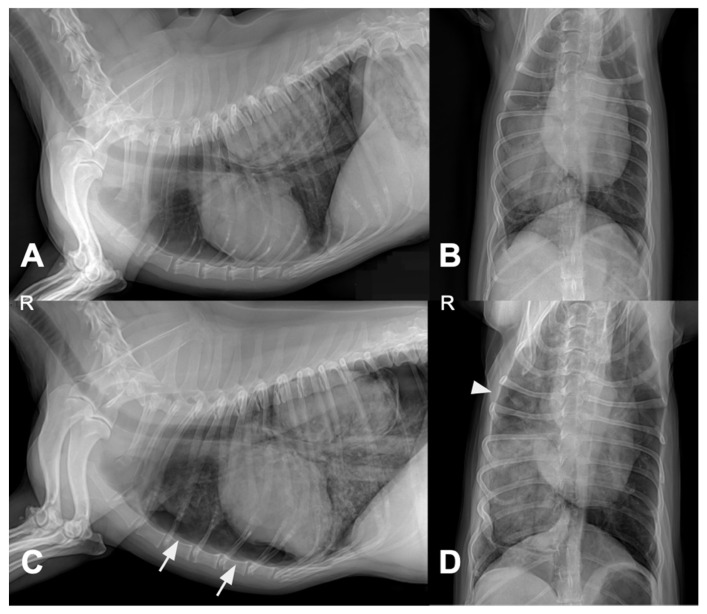
Thoracic radiographs before (**A**,**B**) and four days after (**C**,**D**) vinorelbine administration and fine-needle aspiration. Pretreatment images (**A**,**B**) show a solitary pulmonary mass without notable perilesional changes. Follow-up images (**C**,**D**) demonstrate new abnormalities, including increased alveolar opacity, pneumothorax (arrow in (**C**)), and mild pleural effusion (arrowhead in (**D**)). Abbreviations: R, right side.

**Figure 5 animals-15-03330-f005:**
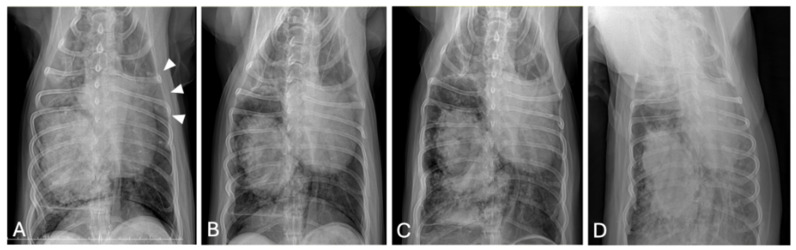
Ventrodorsal thoracic radiographs at selected key follow-up time points. (**A**) Week 14: A new metastatic lesion is present in the contralateral lung (arrowheads). (**B**) Week 24–30: No measurable change in the primary pulmonary mass compared with previous imaging. (**C**) Week 30: An increase in the size of the primary pulmonary mass is observed compared with prior examinations. (**D**) Week 34: A hyperlucent area within the mass and mediastinal shift with compression of the contralateral lung are evident.

## Data Availability

The data supporting the findings of this study are available from the corresponding author upon reasonable request.

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
