# Peer review of "A Case Report of Advanced Pulmonary Adenocarcinoma in a Dog Managed with Chemotherapy and Cytokine-Based Immunotherapy"

_animals, 2025, doi:10.3390/ani15223330_

Round 1
Reviewer 1 Report
Comments and Suggestions for Authors
Dear authors,
The case report is potentially interesting, but it has been written with little care and that there are gaps in the information that should be highlighted more clearly in the article.
-Simple summary: The fact that there is a myelolipoma in the spleen does not seem interesting enough to be included in the simple summary. I would explain more clearly that the pulmonary diagnosis was based solely on cytology, that no lymph node cytology was performed (at least according to the article), and I would describe the progression of the disease in more detail.
-The introduction is very brief; not everyone is informed about the clinical stage and related survival rates in dogs. I want a broader overview of the topic in general, the stage, and the use of immunotherapies in dogs.
-line 74 add suspected metastasis to the middle ....
-line 75-75: explain what it means to see strong rim contrast enhancement. Is this indicative of metastasis? If so, please include this in the text and explain it. I am not a radiologist, and it is not clear to me.
--89-96 Why do you describe the results in the figure captions and not in the text? This error is repeated throughout the article. The results should be in the text, and the captions should describe the specific photo. Furthermore, it cannot be a hematoxylin and eosin stain, since it is a cytological examination.
-Figure 3: Much of the clinical history is described in the caption, where only the timeline of the case should be included. Furthermore, line 118 does not specify how the new lesion was identified, and this is revealed only later in the article.
-Figure 5: Please double-check as D and E are incorrect; C and D are correct. The same applies to lines 161, 169, and 173. Please double-check everything.
-Discussion, from 190 to 199: discuss the prognosis, but I would not be so sure about the presence or absence of lymph node metastasis, as no cytology or histology of the lymph node was performed. This is undoubtedly a limitation of the article that should be discussed.
-The section on emphysema is also very long in the discussion. I would argue further about the article's limitations, such as the lack of histology, the lesion's histological grade, and cytology and lymph node histology.
-lines 275 and 276: I would point out to the authors that they should have checked the article much more carefully, as there are superficial errors that indicate that it has not been proofread carefully, for example, line 285 PA instead of PAC, and I do not understand the italics on line 286.
Reviewer 2 Report
Comments and Suggestions for Authors
Title: A Case Report of Advanced Pulmonary Adenocarcinoma in a Dog Managed with Chemotherapy and Cytokine-Based Immunotherapy
Comments:
In this manuscript, the authors presented the prospective clinical benefits of combining vinorelbine (a chemotherapeutic agent) with NK cell–activating cytokines (IL-15, IL-12, IL-23, and selenium) in a dog diagnosed with stage IV pulmonary adenocarcinoma. The presented method is an alternative to the traditional chemotherapy, which usually implies high doses that often trigger a number of severe side effects, some of them necessitating hospitalization. Consequently, cytokine-based NK cell immunotherapy combined with chemotherapy could become a desired option, especially for individuals bearing inoperable tumors or, why not, for cases with associated renal or hepatic dysfunction. Since it is a relatively novel concept for veterinary oncology, its great promise still requires further scientific investigation to validate its efficacy. Since canine tumors are frequently diagnosed, in some circumstances with a poor prognosis, the presented case report is important in the field, and the assessed paper may be of great importance for veterinary oncology. Furthermore, the existing therapeutic protocols necessitate continuous improvement to increase the survival period of affected subjects. Overall, the paper is well-documented and written, and the results are clearly presented. Still, the authors should be attentive to the details presented below in order to have a better presentation of the material:
- Page 1 – Abstract (lines 32-35): The last part of the sentence that follows is strange: ”The dog presented with intermittent coughing, dyspnea, and cyanosis, and imaging revealed a large pulmonary mass with suspected nodal metastasis (stage III, T4N1M0). Cytology confirmed pulmonary adenocarcinoma, and a concurrent splenic myelolipoma was surgically removed. Cytology confirmed pulmonary adenocarcinoma, and a concurrent splenic myelolipoma was surgically removed.” In the suggested context, the terminal part of the last sentence sounds unusual. In fact, the awkward link between the lung tumor and splenic tumor…. The authors should split the details related to the lung tumor from the splenic tumor. Rephrase it, please.
- It would be interesting for the authors to mention why they chose, in the proposed study, a primary adenocarcinoma that is occasionally diagnosed in dogs as compared to humans. Why not some other, more common, tumor type? Please include this in the body text of the manuscript.
- Caption details of Figure 1: In between different sentences, punctuation items are missing, making the text hard to understand.
- Fig 2: The acinar feature should be pointed out by the authors in the microphotographs. However, the extensive description of the cytological features should be included in the body text of the manuscript rather than in the figure caption. In the figure caption, the authors should only highlight (including by arrows) the main cytological peculiarities of the neoplastic cell.
- How do the authors calculate the dose for the cytokine? Based on some other reports? This information could be introduced in the paper, too.
Comments on the Quality of English LanguagePlease verify in the comments.
Round 2
Reviewer 1 Report
Comments and Suggestions for Authors
Dear authors, thank you for following my suggestions. In this form, the manuscript seems to me much improved and suitable for publication.